# Antimicrobial Resistance in a Tertiary Care Hospital in Armenia: 2016–2019

**DOI:** 10.3390/tropicalmed6010031

**Published:** 2021-03-07

**Authors:** Hayk Davtyan, Ruzanna Grigoryan, Lyudmila Niazyan, Mher Davidyants, Tehmine Ghalechyan, Karapet Davtyan

**Affiliations:** 1Tuberculosis Research and Prevention Center NGO, Yerevan 0014, Armenia; haykdav@gmail.com (H.D.); ruzanna.grigory@gmail.com (R.G.); 2Nork Infection Clinical Hospital of Ministry of Health of Armenia, Yerevan 0047, Armenia; lyudmila.niazyan@gmail.com (L.N.); davidyants@gmail.com (M.D.); t.ghalechyan88@gmail.com (T.G.)

**Keywords:** antimicrobial resistance, drug resistance, operational research, stewardship, risk factors, surveillance, International Health Regulations

## Abstract

Antimicrobial resistance (AMR) is the acquired ability of pathogens to withstand antimicrobial treatment. To bridge the gap in knowledge for implementing effective and targeted interventions in relation to the AMR in Armenia, we designed this study to explore the performance of AMR diagnostics and the profile of AMR in the Nork Infection Clinical Hospital (NICH) for the period of 2016–2019, particularly to (i) determine the proportions of antimicrobial resistance among all samples tested at the hospital laboratory, (ii) determine the proportion of resistance against specific antimicrobials, and (iii) identify factors associated with AMR. A cross-sectional study was conducted with a secondary data analysis that included all the patients tested for AMR in the laboratory of the NICH for the period of 2016–2019. For this period, only 107 (0.3%) patients out of 36,528 had their AMR test results available and of them, 87 (81%) had resistance at least to one tested antimicrobial. This study has provided some valuable information on the AMR situation in Armenia. The results call for immediate actions to control the access to and the use of antimicrobials, strengthen AMR surveillance, and improve laboratory capacity for the proper and fast identification of drug resistance through a comprehensive system.

## 1. Introduction

Antimicrobial resistance (AMR) is the acquired ability of pathogens—whether they be viruses, bacteria, parasites, or fungi—to withstand antimicrobial treatment. This ability is acquired rather than innate and is secondary to inadequate exposure to treatment, which provides time for the pathogen to mutate and develop a mechanism to resist pharmacological therapy. However, the transfer of the resistant strains is possible, and patients can get a resistant infection from others. AMR leads to ineffective standard therapies, which results in longer and more complicated treatment courses, requiring additional tests and more expensive treatment regimens [1,2]. As a result, AMR is associated not only with increased burdens of morbidity and mortality, but also with increased cost to health care systems.

The genetic changes that result in AMR are a part of a natural process that is constantly occurring—however the process is accelerated when antimicrobials are misused. Misuse can take many forms, such as incomplete treatment courses, overuse of medicines when not indicated, poor or non-existent infection prevention and control programs, poor laboratory capacity, and inadequate surveillance and regulation [2,3]. AMR is a global issue that can be seen in every country at every level of care, and is particularly seen with therapies targeting tuberculosis (TB), malaria, HIV, influenza, pneumonia, and gonorrhea [2].

Armenia is not an exception. The purchase of antimicrobial drugs without a doctor’s prescription is highly prevalent within the country. In 2015 the Armenian government implemented an AMR strategy and action plan in line with World Health Organization (WHO) global efforts to tackle AMR [4,5]. This push has led to a greater understanding of AMR in TB and HIV, however a comprehensive analysis of the situation of AMR within Armenia is still lacking.

While antimicrobial resistance is poorly understood in Armenia, several studies have attempted to assess the rates of self-medication in the country. In 2005, the prevalence of intended self-medication and the actual prevalence of self-medication were found to be 53.1% and 12.5%, respectively [6]. In 2016 the prevalence of self-medication was found to be 26%, of which only 43% used antibiotics in a proper manner. In addition, 54.4% of the study participants reported that antibiotics can help treat the common cold, and 43% reported that antibiotics can be effective against viruses [5].

The present available evidence indicates that the AMR is a major public health problem within Armenia, however little research exists for public policy to tackle the issue. In particular, the assessment of the performance of AMR diagnostics and the profile of AMR in the main facilities in charge of controlling infectious diseases in Armenia can help bridge the gap in knowledge in order to implement effective and targeted interventions. Thus, we designed this study to explore the performance of AMR diagnostics and the profile of AMR in the Nork Infection Clinical Hospital (NICH) in Armenia for the period of 2016–2019. The specific objectives of this study were to (i) determine the proportions of antimicrobial resistance among all samples tested at the hospital laboratory, (ii) determine resistance profile against specific antimicrobials, and (iii) identify factors associated with AMR.

## 2. Materials and Methods

### 2.1. Study Design

This was a cross-sectional study with a secondary data analysis that included all the patients tested for AMR in the laboratory of the NICH for the period of 2016–2019.

### 2.2. Study Settings

Armenia is a land-locked country in the South Caucuses region with a population of about three million. There are 11 regions in the country and the capital city Yerevan hosts around 1/3 of the total population. About 30% of the population are below poverty threshold, while 95% of the population are literate.

The national strategy for AMR surveillance and prevention was approved by the government of Armenia in 2015. The strategy includes seven directives, as follows: (1) the creation of management mechanisms for the use and consumption of antimicrobial agents; (2) the creation of an epidemiological surveillance system for AMR; (3) the enhancement of infection control measures and of the information systems for monitoring and controlling AMR at a health facility level; (4) the enhancement of AMR testing capacity; (5) the improvement of laws and regulations related to medication use and selling; (6) the raising of awareness within the general public on AMR and the provision of continuous education on AMR for health professionals; (7) the prevention of AMR formation in agricultural settings [4]. While these directions set high-level goals to tackle AMR there is a need for more detailed guidelines containing specific actions to move the country towards achieving the seven goals of this strategy.

With regard to the treatment of infectious diseases, the NICH is the only tertiary level hospital specialized in infectious diseases. All potentially dangerous infections and most of the severe cases of infections are referred (or directly brought to by ambulances) to this hospital for treatment.

Most of the services provided in the hospital are covered by the state budget, while there are some services for which patient can pay (out of pocket payments). AMR testing is done using the standard Kirby–Bauer disk diffusion technique. Broad spectrum antimicrobials are prescribed usually at the beginning of the treatment. If the resistance to specific antimicrobials is identified during the treatment, then those antimicrobials are discontinued and replaced with an available substitute. The list of antimicrobials covered by the state budget that can be used for treating infections at the NICH is extremely limited. Thus, the majority of patients receive broad spectrum antimicrobials available on the list without the results of AMR testing. This approach creates a situation where use of an ineffective antibiotic (misuse) can increase the risk of AMR [3,7].

The practice described above has been in place for decades and potentially adds to the uncontrolled selling of antimicrobials (without prescription), thus there may already be a critical situation related to AMR in the country. The antimicrobial medication market is not well regulated. After registering the antimicrobials in the country, pharmacies are allowed to sell them without prescription. Pharmacists may suggest these medications to potential patients without consulting with a doctor or testing.

### 2.3. Study Population, Data Sources and Variables

The study population included all hospitalized patients who received treatment at the NICH for the period of 2016–2019 and were tested for AMR in the laboratory of the hospital. A database was developed based on the review of the available medical forms and data relevant for this study. Anonymized data was directly abstracted and entered into the database using an online google form linked with the database. Data variables included basic demographics (age, gender, location), and clinical data including initial and final diagnosis, sample type, infectious agent identified, individual sensitivity to 22 antimicrobials, and antimicrobials used for the treatment. The working definition of AMR in this study was the following: AMR is the ability of pathogens to withstand antimicrobial treatment identified by the laboratory testing at the NICH laboratory.

### 2.4. Statistical Analysis

EasyStat online statistical platform (https://easystat.app/, accessed on 9 February 2021), developed on the basis of R statistical software, was used for the final analysis [8]. Descriptive statistics, which included frequencies, means, medians, and standard deviations (SDs), was used to explore the characteristics of the study population; the Chi-square test and t-test were used to test the differences in proportions and means in different groups, respectively. The level of significance was set at 5%. Simple and multiple logistic regression were used to explore the impact of potential risk/protective factors for developing AMR. All collected variables were included in the multiple logistic regression (multivariate) analysis. Multiple models were explored to identify best fit based on the Akaike information criterion.

## 3. Results

For the period of 2016–2019 there were only 107 (0.3%) patients out of 36,528 in total with available AMR test results and all of them were included in the study. The population was very diverse. The geography of patients covered the whole country, with the majority being from the capital region (37%). The average age of the patients was 18.7 years (SD, 21.7) and ranged from 0 to 78 years old with a skewed distribution. The proportion of males was 63%. Out of eight different types of tested specimens, urine was the most common (33%) followed by blood (21%). The demographic and clinical data of the study population are summarized in more detail in Table 1.

The identified pathogens were diverse (16 different species), with *Escherichia Coli* (*E. Coli*) (22%) being the most prevalent followed by *Staphylococcus Aureus* (*S. Aureus*) (18%).

Patients had more than 50 different diagnoses or a combination of diagnoses.

Only 20 (19%) out of 107 tested samples had no resistance, while 87 (81%) had resistance at least to one tested antimicrobial.

The resistance profiles/proportions of the specific antimicrobials are presented in Table 2. For all the antimicrobials tested there was at least one specimen with AMR. For the tested antibiotics the percentage of AMR ranged from 7% (Moxifloxacin) to 83% (Chloramphenicol). However, it should be noted that the number of tests for some of these were very low (minimum six samples). For the antifungals the situation was similar, with AMR ranging from 7% (Itraconazole) to 88% (Nystatin).

No risk factors were found to be associated with the AMR in a non-adjusted analysis. As presented in Table 1. none of the associations of the independent factors with AMR were statistically significant (*p* values > 0.2). However, the risk factor analysis was indefinite due to the small number of sensitive samples (*n* = 20, 19%). This disaggregation by different factors and categories resulted in small observations, which made an adjusted analysis not possible. The performed analysis yielded errors or indefinite results in the statistical software. Thus, the risk factor analysis needs to be repeated in future studies with larger samples.

## 4. Discussion

This is the first study in Armenia to explore the situation related to AMR. The study revealed some disturbing findings.

First, only 0.3% of patients had AMR testing results available. This is a very alarming situation because as the vast majority of patients at the NICH do not have results of AMR testing, they may be receiving antibiotics that are not effective against potentially resistant infections. As an example, there were only 12 feces samples tested during the four year study period, while more than 400 cases of *shigellosis* and intestinal *salmonellosis* is reported annually at the NICH. Most of these are patients receiving antimicrobial treatment. It is unclear how the selection of the patients who need AMR test results is carried out. This lack of testing may be the result of not having national treatment guidelines or having inadequate laboratory testing capacity. In latter case, it is likely that those tested for AMR and thus included in our study were hard-to-treat patients. Among all (107) specimens tested during the period of 2016 to 2019, 81% were resistant to at least one antimicrobial. A study from Israel that covered the period from 2014 to 2017 showed similar findings. In this study, the percentage of carbapenem-resistant *Acinetobacter baumannii* (*A. baumannii*) was 75–80% for different years, while the percentage of resistance for other species ranged from 0% to 32% [9]. A similar picture was observed in South Korea, with 92% of *A. baumannii* being resistant [10]. Resistance seems to be increasing in low- and middle-income countries across the world and not only in humans but also in animals, with the largest hotspots of resistance being located in China and India and new hotspots emerging in Brazil and Kenya [11,12]. It is important to mention that *S. Aureus* was the second most identified pathogen, but we do not have data regarding *Methicillin Resistant S. Aureus* (MRSA). The population-weighted average of the MRSA percentage was 15.5% in 2019 for the countries in the European Union/European Economic Area region [13].

Second, the proportion of AMR for the tested antibiotics ranged from 7% for Moxifloxacin to 83% for Chloramphenicol, making the latter almost unusable. Antifungals showed a similar pattern, with AMR ranging from 7% for Itraconazole to 88% for Nystatin. The continuation of the misuse of antimicrobials can be extremely dangerous and create a situation whereby many antibiotics can become ineffective. The percentage of resistance for some of the widely used, broad-spectrum antibiotics was around 40–50%, for example as seen with Amoxicillin with Clavulanic Acid and Cotrimoxazole. The resistance proportion was around 25% for cephalosporines. According to the study in Israel the proportion of resistant strains for specific antibiotics varied depending on the pathogens. The proportion for Amoxicillin with Clavulanic Acid was around 37% to 50% for *E. coli* and *Klebsiella pneumoniae* (*K. pneumoniae*), while for Cotrimoxazole the resistance ranged from around 2% for *S. Aureus* to 90% for *A. baumannii*. While these resistance patterns have some similarities, for cephalosporines the resistance pattern was worse in Israel, ranging from 10% for *Pseudomonas aeruginosa* (*P. aeruginosa*) to 98% for *A. baumannii* [9].

Third, the risk factor analysis in our study did not yield any significant results due to the small size and diversity of the sample. In the literature there are many known risk factors for developing AMR which may be different depending on the study context, species, and the diseases they cause. According to the WHO, the main risk factors include the misuse and overuse of antimicrobials; inadequate infection control in health care facilities and farms; a lack of access to clean water, sanitation, and hygiene for both humans and animals; low awareness and knowledge; a lack of access to medicines, vaccines, and diagnostics; and a lack of enforcement of legislation [7,14]. In the studies involving animals and farms, the factors found to be associated with AMR included the production size of farms, space, distance from other farms, route of antibiotic administration, and even season [15]. In a systematic review and meta-analysis that explored the carriage of resistant strains of *E. coli,* factors associated with AMR were travel to India and Southeast Asia, diarrhea, and being on a vegetarian diet [16]. Another study that explored AMR in patients with bloodstream infections with *P. aeruginosa* identified factors such as the prior use of antibiotics (fluoroquinolones, carbapenems, or broad spectrum cephalosporins), indwelling urinary catheters, and prior invasive procedures [17].

The strength of this study was the conduct and reporting of the study using the strengthening the reporting of observational studies in epidemiology (STROBE) statement [18]. There were several limitations. The major limitation was the small size and diversity of sample, which resulted in the decreased statistical power to conduct a risk factor analysis including an adjusted analysis. An overly broad AMR definition was used in the study. According to the definition, minor AMR (i.e., to 1st generation cephalosporines etc.), or more severe AMR (i.e., to fluoroquinolones etc.), were both considered to be the same, however these two have a very distinct clinical significance. There may be a potential overestimation of the real resistance levels, as we suspect that data in our sample may belong to hard-to-treat patients with resistant strains of infections. Additionally, we had no control over the collection of the original data which may have contained errors and biases that could impact our analysis.

Even with limitations, our study sheds light on a major public health issue that Armenia is facing and there is an implication. The practice of the uncontrolled sale and misuse of antibiotics has resulted in a drastic situation related to AMR in Armenia. The findings call for as immediate control of the sale and use of antibiotics, to stop resistance formation or at least decrease its speed. One of the key approaches to optimize the use of antimicrobials is the implementation of antibiotic/antimicrobial stewardship programs (ASP) in hospitals. Stewardship not only requires the use of the appropriate drug, but also timely and appropriate dosing for the proper treatment of infections. To be effective, such a system requires an individualized patient approach using a complex dataset and precision antibiotics for personalized treatment. Factors, such as poor integration of such software with medical records, represent common barriers for the effective performance of such systems. Additionally, the benefits of rapid susceptibility testing in hours and not days requires close collaboration between the microbiology laboratory, physicians, and the wider ASP team to ensure timely action [19,20,21,22].

The available data leads us to the conclusion that there is a substantial prevalence and significant burden of AMR in Armenia. This data gives us valuable information to confirm our hypothesis that the human diseases caused by most AMR pathogens are probably undiagnosed and significantly under-recognized, hence their public health importance is vastly underestimated. It creates a crucial need to strengthen AMR surveillance, increase awareness among health care workers to enable them to detect AMR pathogens in a timely manner, improve laboratory capacity to accurately identify the spectrum of drug resistance, and support accurate treatment. National guideline development may be required to set standards of care and ensure the proper diagnosis and treatment of infections and prevent the spread of AMR.

Additionally, these infections may spread to other countries through migration and the short-term travel patterns of patients and carriers. Thus, to prevent and control outbreaks of resistant infections in Armenia, as well as to limit the pathogens’ spread to other countries, it is of critical importance for the country to have an infrastructure for accurately and rapidly identifying AMR pathogens of public health importance including in animals and other environmental sources.

There is a need to have a comprehensive system to monitor antibiotic resistance and develop effective reporting at a national as well as an international level, for quick decision making and in order to comply with the International Health Regulations.

## 5. Conclusions

This study has provided some valuable information on the situation related to AMR in Armenia. The proportion of resistance to at least one antimicrobial in the samples is as high as 81% among those tested, while the proportion of testing remains very low. This calls for immediate actions to control the access to and use of antimicrobials, strengthen AMR surveillance, increase awareness among health care workers to enable them to detect AMR pathogens in a timely manner, and improve laboratory capacity for the proper and fast identification of drug resistance.

## Figures and Tables

**Table 1 tropicalmed-06-00031-t001:** Descriptive demographic and clinical data of the study population.

Characteristics		Total, *n* = 107		AMR Detected				*p*-Value
				Yes, *n* = 87		No, *n* = 20		
		*n*	(SD/%)	*n*	(SD/%)	*n*	(SD/%)	
Age		19	(22)	19	(22)	18	(19)	0.8 ***
Gender	Male	67	(63)	54	(81)	13	(19)	
	Female	40	(37)	33	(83)	7	(18)	0.8 *
Visit Type	Primary Visit	53	(50)	43	(81)	10	(19)	
	Referred	53	(50)	43	(81)	10	(19)	1 *
	Missing	1	(1)	1	(100)	0	(0)	-
Outcome	Cured or Improved	93	(87)	74	(80)	19	(20)	
	No Improvement or Aggravated	14	(13)	13	(93)	1	(7)	0.5 **
Region	Yerevan/Capital City	40	(37)	34	(85)	6	(15)	
	Region	67	(63)	53	(79)	14	(21)	0.5 *
Comorbidities	No	59	(55)	45	(76)	14	(24)	
	Yes	42	(39)	36	(86)	6	(14)	0.2 *
	Missing	6	(6)	6	(100)	0	(0)	-
Types of Specimen	Feces	12	(11)	11	(92)	1	(8)	
	Blood	23	(22)	18	(78)	5	(22)	0.6 **
	Swab (throat)	4	(4)	3	(75)	1	(25)	0.5 **
	Sputum	17	(16)	14	(82)	3	(18)	0.6 **
	Swab (vaginal)	1	(1)	1	(100)	0	(0)	1 **
	Urine	35	(33)	31	(89)	4	(11)	1 **
	Spinal fluid	2	(2)	2	(100)	0	(0)	1 **
	Swab (wound)	6	(6)	4	(67)	2	(33)	0.3 *
	Missing	7	(7)	3	(43)	4	(57)	-
Pathogen	*Escherichia coli*	23	(21)	19	(83)	4	(17)	
	*Candida albicans*	1	(1)	0	(0)	1	(100)	-
	*Enterococcus* spp.	6	(6)	5	(83)	1	(17)	1 **
	*Aspergillus fumigatus*	8	(7)	8	(100)	0	(0)	-
	*Klebsiella pneumoniae*	2	(2)	2	(100)	0	(0)	-
	*Proteus mirabilis*	2	(2)	2	(100)	0	(0)	-
	*Proteus vulgaris*	2	(2)	2	(100)	0	(0)	-
	*Pseudomonas aeruginosa*	8	(7)	7	(88)	1	(13)	1 **
	*Salmonella* spp.	5	(5)	3	(60)	2	(40)	0.3 **
	*Shigella* spp.	3	(3)	2	(67)	1	(33)	0.5 **
	*Staphylococcus aureus*	19	(18)	13	(68)	6	(32)	0.5 **
	*Staphylococcus epidermidis*	11	(10)	10	(91)	1	(9)	0.6 **
	*Streptococcus group A*	4	(4)	3	(75)	1	(25)	1 **
	*Streptococcus pneumoniae*	8	(7)	7	(88)	1	(13)	1 **
	*Yersinia pseudotuberculosis*	1	(1)	1	(100)	0	(0)	-
	Zygomycetes	3	(3)	3	(100)	0	(0)	-
	Missing	1	(1)	0	(0)	1	(100)	-
Antimicrobial Adjustment Based on the AMR Test Results	No	91	(85)	76	(84)	15	(16)	
	Yes	16	(15)	11	(69)	5	(31)	0.2 *

* Chi square test; ** Fisher exact test; *** Student *t*-test. AMR: antimicrobial resistance; SD: standard deviation.

**Table 2 tropicalmed-06-00031-t002:** Proportion of resistance of specific antimicrobials.

Antimicrobials	Total Samples	Resistant Samples	% of Resistance *
Ampicillin	86	36	42%
Amoxicillin/Clavulanic Acid	87	32	37%
Cefuroxime	82	21	26%
Ceftriaxone	90	19	21%
Cefotaxime	21	5	24%
Ceftazidime	48	12	25%
Erythromycin	37	18	49%
Gentamycin	46	8	17%
Amikacin	6	3	50%
Doxycycline	81	24	30%
Chloramphenicol	6	5	83%
Moxifloxacin	70	5	7%
Ciprofloxacin	91	14	15%
Nitrofurantoin	34	4	12%
Cotrimoxazole	67	32	48%
Imipenem	11	3	27%
Nystatin	8	7	88%
Ketoconazole	14	8	57%
Fluconazole	14	8	57%
Itraconazole	14	1	7%
Voriconazole	12	9	75%

* Overestimation of the % of resistance is possible due to selection bias.

## Data Availability

Data is available upon request.

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
