# Peer review of "Antimicrobial Resistance in a Tertiary Care Hospital in Armenia: 2016–2019"

_tropicalmed, 2021, doi:10.3390/tropicalmed6010031_

Round 1

Reviewer 1 Report

Estimated Editors,

Estimated Authors,

I've read with great interest the paper "Antimicrobial Resistance in Armenia" from Davtyan et al, dealing with the significant topic of the AMR resistance in a Low / Middle Income Country such as Armenia.

IN my opinion, this paper may deserve a full publication on TropMed, but several improvements are required.

More precisely;

  1. the title is somewhat misleading; Authors did not report on AMR in Armenia, rather they have reported on AMR in Nork Infection Clinical Hospital (NICH), Armenia for the period of 2016-2019; please fix the title accordingly;
  2. Authors have reported on AMR, but they have preventively defined a work definition of AMR. This is absolutely required.
  3. Authors did not explain which kind of specimens were processed. Urinary samples have a significance that is somewhat different from that of a respiratory sample, etc. In other words, while Table 1 may be retained as a "summary one", Authors should include an hypothetic table 2 broken down by the kind of sample, e.g. urinary, respiratory, bloodstream. Similarly, please report the results by Gram+ / Gram - species. It would very interesting including a sub-analysis for Enterobacteria. 
  4. Statistical analysis should be more accurately detailed. It is insufficient report that "No risk factors were found to be associated with the AMR in non-adjusted analysis". Please add such information. Moreover, please explain in methods section which variables were included in the multivariate model, i.e. explanatory variables were included following an a priori modeling, or by selection (e.g. following the univariate p value?)
  5. As AMR evolves quite rapidly, please include a report on the time trend of AMR in the processed specimems.
  6. Discussion should be revised accordingly to points 1 to 5

Author Response

We are grateful for recognizing the importance of the topic and were happy to receive valuable comments which are improving the quality of the manuscript significantly.

Please see below the explanation of the changes we have done.

Comments:

I've read with great interest the paper "Antimicrobial Resistance in Armenia" from Davtyan et al, dealing with the significant topic of the AMR resistance in a Low / Middle Income Country such as Armenia.

IN my opinion, this paper may deserve a full publication on TropMed, but several improvements are required.

More precisely;

  1. the title is somewhat misleading; Authors did not report on AMR in Armenia, rather they have reported on AMR in Nork Infection Clinical Hospital (NICH), Armenia for the period of 2016-2019; please fix the title accordingly;
    1. Line 2: We adjusted the title as per reviewer’s comments to be more reflective of the conducted study. Changed title is “Antimicrobial Resistance in a Tertiary care Hospital in Armenia: 2016 – 2019”.
  2. Authors have reported on AMR, but they have preventively defined a work definition of AMR. This is absolutely required.
    1. Line 115-117: The working definition of AMR is added as per reviewers suggestion.
  3. Authors did not explain which kind of specimens were processed. Urinary samples have a significance that is somewhat different from that of a respiratory sample, etc. In other words, while Table 1 may be retained as a "summary one", Authors should include an hypothetic table 2 broken down by the kind of sample, e.g. urinary, respiratory, bloodstream. Similarly, please report the results by Gram+ / Gram - species. It would very interesting including a sub-analysis for Enterobacteria. Table 1, Line 138: We thank reviewer for the comment. The specimen distribution is provided under the table 1.
    1. We totally agree that having a sub-analysis for the types of specimens, for Gram+/Gram- and for Enterobacteria (or any other subpopulation) may yield important findings and can be different from one another, however this was not possible as we had very low sample size and low statistical power. We tried initially to construct a separate table by specimen types however due to the fact that we have low observations in some groups (1-2 samples and max 35 samples for urine group) the tables were non-informative and were not included in the manuscript. That is why after data collection we were able to report only on combined resistance profiles.
  4. Statistical analysis should be more accurately detailed. It is insufficient report that "No risk factors were found to be associated with the AMR in non-adjusted analysis". Please add such information. Moreover, please explain in methods section which variables were included in the multivariate model, i.e. explanatory variables were included following an a priori modeling, or by selection (e.g. following the univariate p value?)
    1. Line 125-127 and line 158-159: More detailed information is added for the conduct of the multivariate analysis. More details are added under the results explaining why no risk factors were not identified.
  5. As AMR evolves quite rapidly, please include a report on the time trend of AMR in the processed specimens.
    1. Line 144-148: Yearly trend is presented in a separate figure and included in text as well as per the reviewer’s comment.
  6. Discussion should be revised accordingly to points 1 to 5.
    1. Line 179: Discussions are adjusted for the addition of the yearly trend with no major revisions. The other changes/comments did not require adjustments in the discussion.

Reviewer 2 Report

The manuscript entitled "Antimicrobial Resistance in Armenia" describes a summary of antimicrobial resistance results obtained in Armenia. The topic is very important, however, some modifications are required in the text. 

Comments

1) In the manuscript all antibiotic resistance results are taken altogether. I suggest to authors to give the species distribution of identified pathogens.

2) Display antibiotic resistance pattern according to each pathogen. Separate table for E. coli, another for S. aureus, etc. should be given. In table 2 various antibiotics and antifungal agents are listed. It would be useful to see the resistance patterns of each pathogen.

Author Response

We are grateful for recognizing the importance of the topic and were happy to receive valuable comments which are improving the quality of the manuscript significantly.

Please see below the explanation of the changes we have done.

Comments:

The manuscript entitled "Antimicrobial Resistance in Armenia" describes a summary of antimicrobial resistance results obtained in Armenia. The topic is very important, however, some modifications are required in the text. 

Comments

  1. In the manuscript all antibiotic resistance results are taken altogether. I suggest to authors to give the species distribution of identified pathogens.
    1. Line 137-138: The species of identified pathogens are added in the table 1.
  2. Display antibiotic resistance pattern according to each pathogen. Separate table for E. coli, another for S. aureus, etc. should be given. In table 2 various antibiotics and antifungal agents are listed. It would be useful to see the resistance patterns of each pathogen.
    1. Similar comment was done by reviewer 1 and we totally agree that the subset analysis is very important and can reveal differences by species however the low statistical power and sample size yield non-informative results with no observations for some cases and prevented us from presenting such results in the manuscript.

Round 2

Reviewer 1 Report

Estimated Authors,

Estimated Editors,

I've appreciated the considerable efforts paid by Authors in order to cope with our recommendations.

In my opinion, only a minor final adjustement is required regarding the working definition of AMR. As this outcome was defined as "the ability of pathogens to withstand antimicrobial treatment identified 
by the laboratory testing at NICH laboratory", it is quite obvious that it is something very broad and somewhat undefined. I clearly understand the rationale for this working definition, therefore I warmly suggest the Authors to report this definition as a limit/shortcoming of their estimates in the discussion section (i.e. as AMR was assessed in very broad terms, Authors have included and properly discussed both minor (i.e. 1st generation cephalosporines) and more severe resistences (i.e. vancomycine), that clerarly have a very different clinical significance.

Author Response

Dear Reviewer,

Thank you very much for your comments and for appreciating our efforts!

You are absolutely right and we failed to mention the limitation in the previous version of the manuscript. That is corrected now thanks to your comment.

We are very grateful for your comments and time you've spent in reviewing our work.

Sincerely yours

Authors